**Effects of Anthropogenic Pollutants on Biogenic Secondary Organic Aerosol Formation in the Atmosphere of Mt. Hua, China**

Can Wu[1,2], Yubao Cheng[1], Yuwei Sun[1], Huijun Zhang[1], Si Zhang[1], Cong Cao[3♣], Jianjun Li[3], Gehui Wang[1,2*]

[1]Key Lab of Geographic Information Science of the Ministry of Education, School of Geographic Sciences, East China Normal University, Shanghai 210062, China
[2] Institute of Eco-Chongming, 3663 North Zhongshan Road, Shanghai 200062, China
[3]State Key Laboratory of Loess and Quaternary Geology, Institute of Earth Environment, Chinese Academy of Sciences, Xi'an 710061, China
♣Now at Department of Chemistry, Hong Kong University of Science & Technology

*Corresponding author. Por. Gehui Wang,
Mailing address: School of Geographic Sciences, East China Normal University, Shanghai 210062, China
E-mail address: ghwang@geo.ecnu.edu.cn, or wanggh@ieecas.cn (Gehui Wang)

**Abstract:** Anthropogenic effects on biogenic secondary organic aerosol (BSOA) formation in the upper boundary layer are still not fully understood. Here, a synchronized 4-hourly monitoring of three typical BSOA tracers from isoprene, monoterpenes, β-caryophyllene and other particulate pollutants was conducted at the mountain foot (MF, 400 m a.s.l.) and mountainside (MS, 1120 m a.s.l.) of Mt. Hua, China, to investigate the chemical evolution of BSOA in air mass lifting. Our findings revealed that BSOA was the predominant source of organic matter (OM) at MS site, with an average fraction of ~43% being ~7-fold of that at MF site. As the prevalent BSOA tracer, the isoprene-derived SOA tracers ($BSOA_I$) maintained comparable level at MF site ($183\pm81$ ng/m$^3$) and MS site ($197\pm127$ ng/m$^3$), yet exhibited an inverse diurnal pattern between both sites. And the $BSOA_I$ fraction in OM aloft moderately decreased during the daytime, and correlated positively with 2-methyltetrols/2-methylglyceric acid ratio but negatively with NOx transported from ground level, indicating that anthropogenic NOx would significantly affect the daytime BSOA formation aloft by inhibiting the $HO_2\cdot$-pathway products. Additionally, the further formation of sulfate in lifting air mass would significantly enhance aerosol water content aloft, which suppressed the reactive uptake of isoprene epoxydiol and ultimately diminished the $BSOA_I$ yields during the daytime. These findings provide more insight into the intricate anthropogenic−biogenic interactions affecting BSOA formation in the upper boundary layer atmosphere.

**Keywords:** Biogenic Secondary Organic Aerosol; Isoprene; Anthropogenic pollutants; Vertical distribution

## 1. Introduction

Volatile organic compounds (VOCs) play a crucial role in atmospheric chemistry (Mcfiggans et al., 2019; Coggon et al., 2021), exerting profound influences on the atmospheric oxidizing capacity, tropospheric ozone burden and regional climate (Mellouki et al., 2015; Wu et al., 2020). Among the diverse VOCs, biogenic VOCs (BVOCs) primarily emitted by terrestrial vegetation predominate the global VOC flux at 1 Pg/yr (Guenther et al., 2012), exceeding anthropogenic sources by an order of magnitude. The high reactivity of dominant BVOCs (particularly, isoprene, monoterpene, and sesquiterpenes) towards $O_3$, $OH\cdot$ and $NO_3\cdot$ would drive rapid formation of secondary organic aerosol (SOA) (Zhang et al., 2018a; Ng et al., 2017). Consequently, these biogenically-derived SOA (BSOA) would represent a prominent contribution to global SOA budget (Kelly et al., 2018; Hodzic et al., 2016), although the models remain highly uncertain in BSOA prediction owning to the complexity of physicochemical processes involved (Hallquist et al., 2009). Given the significant climate interactions and public health implications of BSOA (Scott et al., 2014; Shrivastava et al., 2017), advancing the understanding of BSOA, including its precursors, formation processes, is urgently needed.

Over the past two decades, mounting evidence indicate that anthropogenic pollutants (e.g., $SO_2$, NOx) critically regulate the BSOA formation through altering oxidation pathways and gas-to-particle partitioning processes (Xu et al., 2015; Zheng et al., 2023; Xu et al., 2016). For example, sulfate, acting as an effective nucleophile, greatly promotes ring-opening reactions of isoprene epoxydiols (IEPOX) and

subsequent SOA formation, particularly organosulfates and corresponding oligomers (Cooke et al., 2024a; Xu et al., 2015); These products contribute significantly to IEPOX-driven SOA and can alter aerosol physicochemical properties (e.g., phase state, viscosity and morphology) (Lei et al., 2022; Riva et al., 2019), thereby kinetically mediating the reactive uptake of IEPOX, as well as the following SOA yield and evolution (Drozd et al., 2013; Zhang et al., 2018b; Armstrong et al., 2022). Furthermore, a recent laboratory study demonstrates that sulfate/bisulfate equilibrium also plays a critical role in BSOA formation, especially under highly acidic conditions (Cooke et al., 2024b). NOx as an important driver for BSOA formation could alter the fate of organo-peroxy radicals ($RO_2\cdot$), and subsequently affect the yield and chemical composition of BVOC-oxidized products by changing oxidation pathways (Lin et al., 2013; Pye et al., 2010). Specifically, $RO_2\cdot+HO_2\cdot$ reactions under low NOx conditions would predominantly yield low-volatility hydroperoxide species; Conversely, the reaction of NO with $RO_2\cdot$ in the high NOx regime will produce organonitrates and alkoxy radicals ($RO\cdot$) that can fragment into more volatile products. Whereas, the impact of NOx on BSOA yield is nonlinear as explored by laboratory studies (Xu et al., 2014; Kroll et al., 2006). Additionally, the abundant $O_3$ could also significantly promote BSOA formation via enhancing BVOCs ozonolysis. Therefore, the changes of anthropogenic emissions would potentially perturb the BSOA formation. As modeling studies demonstrated, anthropogenic emission controls substantially decreased the BSOA formation in the United States during 1990-2012 (Ridley et al., 2018), and ~ a further 35% of isoprene-derived SOA ($SOA_I$) would be reduced in

2025 if ongoing the similar emission regulations (Marais et al., 2016). The parallel effects also emerged in China, where the effective control on $SO_2$ emissions drove a significant decline of $SOA_I$ at $-8.0\%$/yr over $2007-2015$, even being two-fold that of $SO_4^{2-}$ (Dong et al., 2022).

Numerous studies have comprehensively characterized the surface BSOA, yet the vertical distribution of BSOA remains insufficiently understood, which is a critical driver of uncertainties in the global climate models (Nazarenko et al., 2017; Hodnebrog et al., 2014). The mountain-based observations revealed that BSOA constitutes a substantial fraction (30–60%) of aerosols in the free troposphere (Fu et al., 2014; Ren et al., 2019; Yi et al., 2021); And these elevated BSOA are significantly influenced by the valley breeze that could transport the surface pollutants to the upper boundary layer, indicating the potential effects of surface pollutant emissions on BSOA formation aloft. In addition, airborne pollutants likely undergo aging in the vertical transport process (Wu et al., 2022), causing increasingly complex compositions and changes in oxidizing capacity and aerosol properties compared with that at ground level. Consequently, more observational studies are necessary to obtain an improved understanding of the anthropogenic−biogenic interactions driving BSOA formation aloft.

Guanzhong Basin of inland China is a typical semiarid region in East Asia, suffering serious particle pollution due to the large anthropogenic emissions (Wang et al., 2016). In our previous studies (Li et al., 2013; Wang et al., 2012), the molecular distribution, evolutionary mechanism and sources of BSOA have been investigated;

Whereas the anthropogenic emissions experienced dramatic changes recently in this
region (Zhang et al., 2019a), thereby, the primary factors currently driving the BSOA
formation are probably distinct from those that prevailed previously. To elucidate the
formation mechanism of biogenic SOA aloft in this region, synchronous observations
were conducted on the mountainside and the mountain foot of Mt. Hua, which adjoins
the Guanzhong basin. In this study, we firstly investigate chemical molecular
compositions and diurnal variation of the of BSOA over Mt. Hua, then explored the
impacts of anthropogenic pollutants on BSOA formation during the vertical transport,
and finally quantified its source contributions.
**2 Experiment**
**2.1 Sample collection**
From 27 August to 17 September 2016, aerosol sampling with a 4-hr interval were
synchronously conducted at two locations in Mt. Hua region, employing the high-
volume air samplers with a flow rate of 1.13 $m^3$/min. One sampling site (34°32′N,
110°5′E, 400 m a.s.l; MF) is situated at the mountain foot of Mt. Hua, enveloped by
several traffic arteries, residential and commercial buildings. Another site is located
on mountainside (34°29′N, 110°3′E, 1120 m a.s.l; MS), approximately 8 km away
from MF site in horizontal distance; This site is adjacent to one of the larger valleys of
Mt. Hua, characterized by precipitous terrain and less anthropogenic activities. The
surface-level pollutants can be transported to here by the prevailing valley breeze,
which has been validated by the organic tracers and meteorological field simulated by
WRF-Chem model in our previous studies (Wu et al., 2022; Wu et al., 2024). All the
aerosol samples were collected on pre-combusted (450 °C for 6 hr) quartz filters
(Whatman 1851-865), which has a retention efficiency of >99.995% for DOP
particles at 0.3 µm; After collection, the filter samples would be stored in a freezer (<
−18 °C) prior to chemical analysis.
Additionally, the hourly concentration of the pollutants, including $PM_{2.5}$, $O_3$, $NO_2$,
were also monitored at MS site by corresponding online equipment; While, those data
for MF site were mainly acquired through Weinan Ecological Environment Bureau
(http://sthjj.weinan.gov.cn/, last access: 8 July 2021). All meteorological data were
downloaded from the Shaanxi Meteorological Bureau website (http://sn.cma.gov.cn/,
last access:8 July 2021). The comprehensive details regarding the sampling sites and
instrumentation were delineated in our previous studies (Wu et al., 2022; Wu et al.,

160 2024).

**2.2 Chemical analysis**
The details of organic matter extraction, derivatization, and gas
chromatography/mass spectrometry (GC/MS) analyses can be referred to elsewhere
(Wang and Kawamura, 2005). Briefly, for the analytical procedure of organic tracers
in the aerosol, one-quarter of $PM_{2.5}$ sample was cut into pieces and then ultrasonically
extracted with a mixture of dichloromethane and methanol (2: 1, v/v) three times
(each for 15 min). The extracts were filtered through a pasteur pipette plugged with
quartz wool into a pear-shaped bottle. The filtrates were concentrated by a rotary
evaporator under vacuum state and then dried by pure nitrogen. After reaction with 60
µL    derivatization    reagent    (a    mixture    of    50    µL    of    N,O-bis-
(trimethylsilyl)trifluoroacetamide (BSTFA) and 1 % trimethylsilyl chloride and 10 μL
of pyridine) for 3 h at 70 °C in order to convert COOH and OH groups to the
corresponding trimethylsilyl esters and ethers. After cooling down to room
temperature, an aliquot of 40 μL internal standard ($C_{13}$ n-alkene) was added into the
derivative solution prior to GC/MS analyses. All BSOA tracers were individually
identified by comparing mass spectra against authentic standards and literature data.
Whereas, due to the commercial unavailability of a subset of authentic standards, the
quantification of target compounds relied on the surrogate standards, expect for cis-
pinonic. Specifically, the erythritol was applied to determine the 2-methyltetrols, $C_5$-
alkene triols and 3-MeTHF-3,4-diols; The quantification of 2-methylglyceric acid, 3-
hydroxyglutaric acid, 3-methyl-1,2,3-butanetricarboxylic acid and β-caryophyllinic
acid was performed using glyceric acid, tartaric acid, suberic acid and cis-pinic acid,
respectively. This approach was also applied in other similar studies (Li et al., 2013;
Zhu et al., 2016). The standards solutions were spiked onto the blank filters to assess
to the recoveries, which ranged from 82% to 105%. Field blank filters were also
analyzed following the same analytical procedure as the ambient samples, with no
target compounds detected in these blanks.
Furthermore, partial filters were cut into pieces, and then extracted three times
under sonication with 15ml Milli-Q pure water (18.2 MΩ). Ten ions such as $SO_4^{2-}$,
$NO_3^-$, $Cl^-$, $NH_4^+$, and $K^+$ were determined using ion chromatography (Dionex, ICS-
1100). A DRI model 2001 thermal–optical carbon analyzer was used to measure the
organic carbon (OC) and element carbon (EC) in each $PM_{2.5}$ filter samples, of which

water-soluble organic carbon (WSOC) was extracted into Mill-Q pure water (18.2

MΩ) and ultimately quantified by a total organic carbon (TOC) analyzer (Model

TOC-L CPH, Shimadzu, Japan).

**2.3 Aerosol liquid water content (ALWC) and in situ pH**

The thermodynamic model (ISORROPIA-II) can provide robust predictions of the

aerosol liquid water content (ALWC) and in situ pH in $PM_{2.5}$ samples. By combining

the actual temperature (T) and relative humidity (RH) in the atmosphere and the

composition, the ALWC and $H^+$ loads can be simulated, which are used for

calculating the situ pH.

**2.4 Positive matrix factorization (PMF) source apportionment**

As a receptor model, PMF is a mathematical approach to quantify the contribution

of sources to samples, and has been widely used in the source apportionment of air

pollutant. More details of model can be found on the EPA website

([https://www.epa.gov/air-research/epa-positive-matrix-factorization-50-fundamentals-](https://www.epa.gov/air-research/epa-positive-matrix-factorization-50-fundamentals-and-user-guide)

[and-user-guide](https://www.epa.gov/air-research/epa-positive-matrix-factorization-50-fundamentals-and-user-guide)). In this work, the organic matter (OM), EC WSOC, secondary

inorganic ions, biogenic SOA of all the samples served as the fingerprint species to

identify potential sources of the BSOA at Mt. Hua. After extensive testing, the model

would output an optimal solution characterized by a minimal difference between $Q_{True}$

and $Q_{Robust}$ and a $Q/Q_{expected}$ ratio approaching 1; This is indicative of the robust model

performance, as confirmed by the high correlation coefficients between input and

estimated values. Furthermore, as revealed in our pervious study (Wu et al., 2022), the

change of the main emission sources was insignificant in air mass lifting process on

Mt. Hua. Thus, the samples from both sites were added together as one data matrix.
Based on the Q values ($Q_{True}$: 2362.8 vs. $Q_{Robust}$: 2365.6; $Q/Q_{expected}$: 0.86) and
interpretability, five factors were obtained as the optimal solution after numerous
testes. And the model could commendably reconstruct the temporal profiles of the
input species, showing significant correlations ($R^2 > 0.92$) between observed and
simulated species.
**2.5 Concentration-weighted trajectory (CWT) analysis**
To identify the potential spatial sources of the high BSOA on Mt. Hua, a CWT
analysis was conducted using the BSOA concentrations and air mass backward
trajectories. The 12-hr backward trajectories at a height of 1100 m were calculated by
Hybrid-Single Particle Lagrangian Integrated Trajectory (HYSPLIT) model on an
hourly basis throughout the campaign. Due to the relatively low temporal resolution
of filter samples, the averaged BSOA concentration was assigned to match all the
trajectories corresponding to each sample. And the CWT analysis herein was
subsequently performed using the Igor-based tool using the Igor-based tool (i.e., ZeFir
v3.70); More details for the protocol can can be found in Petit et al. (2017). Briefly, a
grid with 0.25° × 0.25° cell size was created to cover all the trajectories; Each grid
cell would be assigned a weighted concentration, which obtained from the averaged
BSOA concentration that have associated trajectories crossing the grid cell. A high
value in a gird cell indicates that the air parcels passing through it were associated
with high concentration at the receptor site.
**3 Results and discussion**

## 3.1 Overview of OM during the campaign

The bulk chemical compositions of $PM_{2.5}$ during the campaign have been reported in our previous work (Wu et al., 2022), and strongly substantiated that organic matter (OM) is the principal component of $PM_{2.5}$ at both surface and high-elevation sites, with a mass fraction of 32%~46%. For a deeper insight into the relative abundances, vertical variability and sources of OM among two sampling sites, a comprehensive comparison was presented in Figure 1. The OM concentration was $30\pm10$ $\mu g/m^3$ at the surface, approximately three-fold of that aloft. Such abundant OM at surface was apparently driven by fresh emissions from nearby roads and residences, as evidenced by its strong correlation with hopanes (Figure S1(a), $R^2=0.46$, $p<0.05$) being a known tracer for combustion sources (Schauer et al., 1999); Whereas, a weak correlation between OM and hopanes was found at MF site, indicating the different sources for the OM aloft. Moreover, the surface OM was characterized by a higher nocturnal load, and exhibited a decreased trend before midafternoon (Figure S1(b)), which was thermally driven boundary layer growth. Nonetheless, an inverse diel pattern was observed at high elevation site with a daily OM peak at the moment (12:00~16:00) of strong photochemical activity (Figure S1(b)), suggesting that OM components aloft was probably driven by photochemistry.

To determine the sources of OM at both sites, we performed a positive matrix factorization (PMF) analysis, and identified five types of sources for OM (Figure S2). Among these sources, biomass burning (BB) and fossil fuel combustion were believed to be the primary sources for OM at the surface, accounting for 52% and ~18% of

OM (Figure 1c), respectively. This coincided with that in Xi'an (~ 43% for BB) and
Lin Village (30%-40% for BB) located on Guanzhong plain (Elser et al., 2016; Li et
al., 2022). Whereas, merely ~24% of OM was explained by above anthropogenic
sources at MS site; and most of OM aloft was dominated by biogenic secondary
organic aerosol (BSOA, Figure 1d), with a fractional contribution of ~43% being 7-
fold of that for subaerial OM. This elucidated a significant influence of biogenic
primary and secondary sources on OM aloft.
**3.2 Abundance and spatiotemporal variations of BSOA tracers**
To characterize the spatiotemporal variations of biogenic SOA among two sites,
plenty of BSOA tracers including isoprene, monoterpene and sesquiterpene were
quantified as summarized in Table1. The sum of detected BSOA tracers ranged 31
ng/m$^3$ to 459 ng/m$^3$ at MF site (269±108 ng/m$^3$, Figure S3), which was slightly lower
than that at MS site (277±159 ng/m$^3$); Such spatial pattern was also observed at Mt.
Tai on North China Plain (Yi et al., 2021), further underscoring the significance of
biogenic sources in mountainous OM formation aloft. Specifically, isoprene SOA
tracers (BSOA$_I$) as the dominant species accounted for ~70% of the identified BSOA
tracers, with an average of 197±127 ng/m$^3$ at MS site and 183±81 ng/m$^3$ at MF site,
respectively; These were comparable to that in the Kathmandu Valley of India
(183±81 ng/m$^3$) (Wan et al., 2019) and a mid-latitudinal forest zone (~240 ng/m$^3$)
(Zhu et al., 2016), but ~2 orders of magnitude higher than those measured over the
North Pacific (3.6 ng/m$^3$) (Fu et al., 2011). As for total monoterpene SOA tracers
(BSOA$_M$) and sesquiterpene tracers (BSOA$_S$), their abundances were lower by a

factor of approximately 4~6 than BSOA$_I$, probably due to lower emissions of

monoterpene and sesquiterpene from the vegetation in this region. Noteworthily,

BSOA$_M$ exhibited a converse spatial pattern, with a high load at surface (MF: 52±23

ng/m$^3$ vs. MS: 32±20 ng/m$^3$). The coniferous plants, as primary emitter of

monoterpene (Gagan et al., 2023), may be less densely distributed at the ground level

relative to that on Mt. Hua; Thus, relatively abundant BSOA$_M$ at the surface is likely

affect by intensive BB activities that can also release numerous monoterpenes,

thereby potentially augmenting the surface BSOA$_M$ levels. This can be manifested by

that BSOA$_M$ strongly correlated with levoglucosan being known as BB tracer, but

weakly with C$_{29}$-alkane that mainly derived from vegetation emissions (Wang and

Kawamura, 2005) (Figure S4).

**3.2.1 Isoprene SOA tracers**

A total of eight isoprene tracers formed by oxidated reactions of isoprene with NOx

and OH$_2$· were identified in this study, namely C$_5$-alkene triols (cis-2-MTB, trans-2-

MTB and 3-MTB), 2-MGA, 2-methyltetrols (2-MT and 2-ME) and 3-MeTHF-3,4-

diols (trans-3-Methyltetrahydrofuran-3,4-diol and cis-3-Methyltetrahydrofuran-3,4-

diol). The BSOA$_I$ at MF site exhibited a pronounced diurnal cycle, characterized by a

higher daytime load with an afternoon peak of 210±96 ng/m$^3$ (Figure 2(a)). This

pattern aligns with temperature-driven characteristic of isoprene emission (Pétron et

al., 2001; Zeng et al., 2023). A positive correlation between BSOA$_I$ and temperature

(R$^2$=0.63) implies that a substantial part of surface BSOA$_I$ was expected to be formed

locally (Figure S5 (a)), given the short atmospheric lifetime of isoprene. Whereas,

BSOA$_I$ aloft was characterized by an inverse diurnal variation, with a moderate
enhancement in the nocturnal load. As revealed by a high-elevation CWT analysis of
BSOA$_I$ (Figure 3), relatively high nocturnal loading was distributed over Mt. Hua and
adjacent regions, indicating that these SOA tracers aloft were significantly influenced
by regional/super-regional transport, especially during the nighttime.
2-methyltetrols (2-MTLs), as the predominant species at both sites, were more
abundant aloft (99±69 ng/m$^3$) relative to surface site (74±32 ng/m$^3$); The levels were
comparable to the field measurements at Mt. Tai (98 ng/m$^3$) (Yi et al., 2021), Mt.
Changbai (22~282 ng/m$^3$) (Wang et al., 2008) and in Mexico City (8~190 ng/m$^3$)
(Cooke et al., 2024a). Recently, a forest observation demonstrated that 2-MTLs could
be biologically produced and then directly emitted into the atmosphere (Ye et al.,
2021). Whereas, there is no significant correlation between 2-MTLs and C$_{29}$-alkane
mainly released by vegetation (Figure S4(b)), indicating that 2-MTLs at both
sampling sites primarily derived from secondary formation rather than being directly
emitted by vegetation. As the oxidation products of isoprene under low/free-NOx
conditions, 2-MTLs at MF site were significantly formed during the daytime, and
peaked at 12:00~16:00 LT (local time, 85 ng/m$^3$, Figure 2(b)), corresponding to the
period of reduced NOx loads, high temperature and intense solar radiation.
Conversely, the concentration of 2-MTLs aloft decreased progressively during the
daytime, bottoming out at 16:00~20:00 LT; This inverse diurnal pattern was probably
due to the influx of the surface NOx that was transported aloft by the prevailing valley
breeze, which subsequently inhibited the 2-MTLs formation aloft. Meanwhile, such
relatively abundant NOx condition during the daytime would promote the formation
of 2-MGA aloft (culminating at 12:00~16:00 LT), as corroborated by a positive
correlation between 2-MGA and $NO_2$ (Figure S5 (c)). This finding was consistent with
the laboratory measurements of 2-MGA being derived from oxidization of isoprene
with high NOx load (Surratt et al., 2006; Szmigielski et al., 2007). Moreover, the
surface 2-MGA also exhibited a similar diurnal pattern with an average of 29±15
ng/m$^3$ being 1.8-fold of that aloft, albeit a weak relationship between 2-MGA and $NO_2$;
This appreciably suggested that, beyond NOx, other factors also drove the 2-MGA
formation at MF site.
$C_5$-alkene triols and 3-MeTHF-3,4-diols were another prevalent isoprene tracers, of
which concentrations were comparable among two sites (Table 1). As the
photooxidation products under low/free-NOx levels, above two tracers exhibited a
diurnal cycle paralleled to that of 2-MTLs at two sites, and were characterized by
higher loads during nighttime. Despite that, the correlation between $C_5$-alkene triols
and 3-MeTHF-3,4-diols ($R^2$>0.74) was more robust than that involving 2-MTLs. This
was probably owing to that $C_5$-alkene triols and 3-MeTHF-3,4-diols were mainly
formed via acid-catalyzed intermolecular rearrangement reactions of epoxy-diols (Lin
et al., 2012; Wang et al., 2005), whereas 2-MTLs are likely to the result from
nucleophilic addition of water to the ring opening of epoxy-diols (Surratt et al., 2010).
**3.2.2 Monoterpene SOA tracers**
The detected monoterpene tracers, including pinic acid (PA), cis-pinonic acid
(PNA), 3-hydroxyglutaric acid (3-HGA) and 3-methyl-1,2,3-butanetricarboxylic acid

(MBTCA), are mainly derived from photooxidation of α/β pinene initiated by ·OH

and $O_3$; Therefore, those four tracers exhibited a similar diurnal pattern, peaking at

12:00~16:00 LT when ·OH and $O_3$ concentrations remained at highest daily level.

Based on the chamber measurements, PA and PNA are the early-generation products

of α/β pinene (Liu et al., 2022; Jaoui et al., 2005), merely accounting for ~10% and

~12%-17% of the $BSOA_M$, respectively. Strikingly, PNA at two sampling sites was

more abundant by a factor of ~1.5 than PA, consistent with the observations at Mt. Tai,

Mt. Huang and in Duke Forest, North Carolina (Wang et al., 2023; Bhat and Fraser,

2007; Yi et al., 2021). The saturation vapor pressure of PNA (~7.2 × $10^{-5}$ Pa at 298 K,

estimated by the E-AIM model) is lower than that of PA (~2.0× $10^{-4}$ Pa), thus, PNA

can be readily nucleated and saturated in the atmosphere. Given that, PNA was

expected to have a higher abundance in aerosol than PA.

During the campaign, MBTCA was the most abundant $BSOA_M$ tracer, explaining

more than half of the surface $BSOA_M$ tracers at both sites, followed by 3-HGA with

fractional contributions of ~20% to total $BSOA_M$. MBTCA and 3-HGA are regarded

as the later-generation products of α/β-pinene, and can be derived from further

photodegradation of PNA or PA; Thus, MBTCA/(PA+PNA) ratio (abbreviated as M/P

hereafter) is commonly used to estimate the α/β-pinene-derived SOA aging, of which

a higher value is indicative of more aged α/β-pinene SOA. As depicted in Figure S6,

the M/P value (2.6±1.3) at MS site was ~1.4-fold of that at MF site, reflecting that the

$BSOA_M$ tracers aloft were more aged compared to that at surface; This finding

coincided with the variation of oxidation state of carbon (OSc) measured by the

aerosol mass spectrometer. Additionally, a clear diurnal pattern of M/P ratio was found at MF site, with a daily peak at 16:00~20:00 LT lagged that of MBTCA by ~4 hr. However, the diurnal cycle of the M/P ratio at MS site exhibited a bimodal pattern, with two nearly equivalent peaks during the daytime (16:00~20:00) and nighttime (4:00~8:00). Relatively high M/P ratios during nighttime indicated a more aged state of nocturnal α/β-pinene SOA at the upper atmosphere, markedly differing from the ground-based observations. This was partially due to that the nocturnal pollutants at MS site were mostly driven by regional or long-range transport (Figure 3), in which α/β-pinene SOA aloft would undergo a deeper aging, and leading to the high M/P ratio.

### 3.2.3 Sesquiterpene SOA tracers

As a typical and abundant sesquiterpene, β-caryophyllene has been widely studied due to its high reactivity and significant aerosol formation potential; and it can be oxidized into β-caryophyllinic acid via ozonolysis/photooxidation (Jaoui et al., 2007; Jaoui et al., 2003). In this study, β-caryophyllinic acid was found to exhibit a relatively high concentration among the detected BSOA tracers (Table 1), with an average of $35\pm19$ ng/m$^3$ at MF site and $56\pm41$ ng/m$^3$ at MS site, respectively; These loads were comparative to Mt. Wuyi (7.6-54 ng/m$^3$) (Ren et al., 2019) and Mt. Tai (0.05-48 ng/m$^3$) (Fu et al., 2012). As expected, β-caryophyllinic acid robustly correlated with the BSOA$_M$ ($R^2>0.60$, $p<0.05$) at both sampling sites, indicating similar formation pathways for these species. Additionally, the significant relationship between β-caryophyllinic acid and levoglucosan at MF site could be a result of

intensive biomass burning as indicated in section 3.1; This observation aligns with
previous findings that sesquiterpenes accumulated in leaves and wood can be released
in the biomass burning process, subsequently oxidized to form β-caryophyllinic acid
(Zhang et al., 2019b). Notably, β-caryophyllene SOA tracers aloft in this study
showed clearly enhanced concentrations during the daytime from 10:00 to 16:00;
whereas the diurnal cycle of β-caryophyllinic acid at MF site was less pronounced,
despite significant diurnal differences in $O_3$ load and temperature that are known to
influence heterogeneous reactions of β-caryophyllene.

**3.3 Influencing factors of BSOA formation in upper troposphere**

A comparison of diurnal cycles of mass fraction of BSOA to OM ($F_{BSOA/OM}$) among
both sampling sites is illustrated in Figure 4. At MF site, a spike in $F_{BSOA/OM}$ was
observed at around 12:00-16:00 (Figure 4(a)), demonstrating that the surface BSOA
formation was strongly enhanced in the presence of abundant $O_3$/OH radical load and
high biogenic emissions, especially for isoprene SOA; Such diurnal variation was in
good agreement with the measurements at other sites (Zhang et al., 2019b; Zhu et al.,
2016; Lee et al., 2016). Whereas $F_{BSOA/OM}$ for isoprene aloft exhibited a decreasing
pattern during the photochemically active period (Figure 4(b)), reaching a minimum
at 12:00-16:00 LT. Although oxidant levels remained relatively high at this time, the
$F_{BSOA/OM}$-isoprene was only 75% of that for the prior moment. Our previous
observational evidences have corroborated that surface anthropogenic pollutants can
be transported to the upper atmosphere by the prevailing valley breeze (Wu et al.,
2022; Wu et al., 2024). Those anthropogenic pollutants (e.g., $SO_2$, NOx) likely
modulated BSOA formation aloft, leading to above unusual diurnal cycle of $F_{BSOA/OM}$-
isoprene. To elucidate the above hypothesis, the mantel test and random forest (RF)
analysis were applied for daytime BSOA aloft. More descriptions and settings for the
models can be seen in the supplementary materials (Test S1).

**3.3.1 Effects of meteorological factors on BSOA**

As illustrated in Figure 5(a), the daytime $BSOA_I$ aloft positively correlated with
ambient temperature, consistent with the observations on Mt. Tai, Mt. Huang and Mt.
Wuyi (Ren et al., 2019; Wang et al., 2023; Yi et al., 2021). Given the temperature-
driven characteristic of isoprene emission (Pétron et al., 2001), relatively high
temperature was expected to enhance biogenic SOA yield through the photooxidation
of isoprene. While, $BSOA_M$ and $BSOA_S$ exhibited insignificantly negative
correlations with temperature (Figure 5(a)), indicating that their formation may be less
sensitive to temperature variations. Additionally, the strong temperature-dependence
of $BSOA_I$ also indicated a significant part of $BSOA_I$ tracers aloft undergo rapid in-situ
formation once the oxidants concentrations built up during the daytime, contrasting
with those during nighttime.    An insignificant correlation was observed between
relative humidity and BSOA tracers ($|r|<0.08$, $P \geq 0.05$). However, the moist weather
frequently occurred at MS site, even during the daytime with average RH of $62\pm19\%$
(Wu et al., 2022); Thus, the less pronounced variation can explain the insensitivity of
BSOA formation to RH. Even so, the high RH during the daytime could indirectly
influence BSOA formation aloft by modulating aerosol water content (ALWC),
aerosol acidity, gas–particle partitioning of BSOA precursors (Xu et al., 2015; Yan et
al., 2025; Isaacman-Vanwertz et al., 2016).

**3.3.2 Effects of anthropogenic pollutants on BSOA**

Evidence from laboratory studies referred to a nonlinear dependence of BSOA
yield on NOx load, wherein the yield increases with rising NOx levels under low-
NOx conditions, but exhibits a decreasing trend as NOx levels rise under high-NOx
conditions (Xu et al., 2014; Xu et al., 2025). Whereas, the BSOA tracers aloft,
especially isoprene-derived SOA, negatively correlated with $NO_2$ (r=0.36, $P<0.05$)
(Figure 5(a)). Additionally, the 2-MTLs/2-MGA ratio, which is indicative of NOx
influence on SOA formation, followed a similar diurnal pattern to that of the
$F_{BSOA/OM}$-isoprene (Figure 5(b)). All those findings indicated that the $NO_2$ transported
from the surface to high altitude may limit the formation of $SOA_I$ in the daytime. As
is well known, isoprene oxidization primarily follows two pathways, i.e., $HO_2$
pathway forming 2-MTLs in NOx-limited conditions and $NO/NO_2$ channel yielding
2-MGA in high-NOx scenarios (Surratt et al., 2010; Szmigielski et al., 2007). Thereby,
we hypothesized that the increasing NOx at MS site may perturb BSOA formation by
competing with the $HO_2$ pathway, diminishing the net SOA yield relative to NOx-
limited conditions. This was consistent with experimental finding conducted by
Thornton et al. (2020), who demonstrated that at a high NOx level (NO~500 ppt) that
is akin to our daytime observations at MS site, the maximum BSOA yield is 10%
lower than that at low NOx.
As shown in Figure 5(a), the sulfate presented a statistically significant positive
relationship with $BSOA_I$. This feature appeared to be common for other similar field

studies (Wang et al., 2008; Xu et al., 2015; Liu et al., 2017), underscoring the significant role of sulfate in the isoprene-derived SOA formation. According to laboratory studies (Eddingsaas et al., 2010), the sulfate can act as nucleophiles to facilitate the ring-opening reaction of IEPOX, which is a pivotal oxidation product of isoprene when organic peroxy radicals mainly react with $HO_2$ radicals. While, the daytime sulfate level was moderately enhanced, peaking at 12:00-16:00 LT (Figure 5(c)); Additionally, a ~4% enhancement in the mass fraction of daytime sulfate was also observed at MS site compared to MF site (Figure S7(a)). These findings are indicative of the further formation in air mass lifting process (Wu et al., 2022). An increase in sulfate would boost the ionic strength and salting-in effect of aerosol, thereby enhancing IEPOX reactive uptake and inhibiting its reversible partitioning back to the gas phase. The $SO_4^{2-}$/$HSO_4^-$ equilibrium is crucial for $BSOA_I$ yield under a highly acidic condition (Cooke et al., 2024b); While, the chemical form of inorganic S(VI) also shifts from $HSO_4^-$ at MF site to $SO_4^{2-}$ at MS site during vertical transport, which would facilitate $BSOA_I$ formation, as the nucleophilicity of $SO_4^{2-}$ is two orders of magnitude higher than that of $HSO_4^-$ (Aoki et al., 2020). As is well-known, sulfate can modulate the $H^+$ level, acting as a more efficient proton donor to catalyze IEPOX ring opening and isoprene ozonolysis; Thus, $H^+$ concentration predicted by the thermodynamic model ISORROPIA II positively correlates with $BSOA_I$ (r=0.37, *P*<0.05, Figure 5(a)). But a nonlinear relationship between the yields of methyltetrol sulfates/2-MTLs and under low pH (<3) condition is revealed in the chamber study (Cooke et al., 2024b); Based on our previous study (Wu et al., 2022), the average

daytime pH was 3.4±2.2 at MS site, indicating that there are other factors perturbing
BSOA formation at Mt. Hua, warranting further investigation.
Furthermore, the additional sulfate formation in lifting air mass can also promote
an enhancement in ALWC aloft, consistent with the strong positive correlation
observed between ALWC and sulfate ($R^2$=0.66, $P$<0.05, Figure S7). As demonstrated
in Figure 5(c), the daytime ALWC peaked at 12:00-16:00 LT, corresponding to a
significant decrease in $F_{BSOA-isoprene/OM}$. This pattern suggested an inhibited effect of
aerosol water on BSOA formation, consistent with laboratory observation by Gaston
et al. (2014), who found a 50% reduction in IEPOX reactive uptake on $NH_4HSO_4$
particles as RH increases from 30% to 70%. As evidenced by previous studies (Xu et
al., 2015; Riedel et al., 2015), the abundant aerosol water can moderate BSOA
formation by affecting ionic strength, proton donor/nucleophilic activity, and
consequently altering the reactive uptake (e.g., IEPOX) and subsequent reactions. To
quantitatively evaluate the effects of ALWC, the pseudo-first-order heterogeneous
reaction rate constant for IEPOX reactive uptake ($k_{het}$) was calculated for the samples
at MS site following the method of Gaston et al. (2014) (Text S2). The simulated
uptake coefficient of IEPOX ($\gamma_{IEPOX}$) during the daytime was $4.3\times10^{-4}$, being in the
range of field and laboratory observations ($0.1–6.5 \times 10^{-4}$) (Zhang et al., 2017; Gaston
et al., 2014). As shown in Figure 6, the high ALWC commonly corresponds to low
$\gamma_{IEPOX}$ during the daytime, characterized by weak ionic strength and low $H^+_{(aq)}$
concentration. This indicates that enhanced ALWC at MS likely impeded the IEPOX
uptake onto particle surface. Consequently, the average $k_{het}$ at 12:00-16:00 LT was
merely $9.8\times 10^{-8}$ 1/s, which was an order of magnitude lower than that in the rest of
daytime. These results underscore that enhanced ALWC during the daytime would
lead to an insufficiently rapid heterogeneous reaction of IEPOX, and finally
diminished the $BSOA_I$ formation.
To elucidate the key factors that affect BSOA formation aloft, a random forest (RF)
analysis was conducted for the daytime samples at MS site. The RF model
commendably highlighted the significance of the factors contributing to $BSOA_I$
formation, as evidenced by the robust correlations between the predicted and observed
data for both the training and testing datasets ($R^2>0.8$, Table S1), along with minimal
error metrics. As shown in Figure 5(d), the daytime $BSOA_I$ concentration was largely
affected by the ozone (~29%), with a robust positive correlation between them (Figure
5(a)); This indicated that enhanced $O_3$ is likely to augment isoprene oxidation
products aloft. Additionally, $NO_2$ and ALWC also play pivotal roles in $BSOA_I$
formation, with the importance of ~19% and 13%, respectively; These findings
further corroborated that the enhanced NOx may lead to an impedimental effect on the
daytime BSOA formation in the upper boundary layer of Mt. Hua. Such limiting
effect on isoprene-derived SOA formation under a high NOx scenario was also found
in Eastern China and Amazon (Zhang et al., 2017; De Sa et al., 2017).
**4 Summary and conclusion**
The $PM_{2.5}$ samples with 4 h intervals were synchronously collected at mountain
foot and mountainside of Mt. Hua, to elucidate the chemical evolution and
spatiotemporal differences of the organic matter among two sites. At the MF site, the

anthropogenic emissions, such as biomass burning and fossil fuel combustion, were identified as substantial contributors to the OM, accounting for more than 70% of total OM. Whereas, only ~24% of OM aloft was derived from anthropogenic emissions, with biogenic secondary organic aerosol (43%) emerging as the predominant OM source aloft.

Three distinct types of BSOA tracers were identified, predominantly featuring isoprene-derived species. At MF site, most of the BSOA tracers were more abundant during the daytime and peaked at 12:00~16:00 LT, indicative of photochemical oxidation as the primary formation pathway. Whereas, there is a marked decrease in the absolute concentration and relative abundance of daytime isoprene-derived tracers in the upper atmosphere. This decline can be attributed to the intrusion of ground-level NOx, which significantly modifies $BSOA_I$ formation at higher altitudes by inhibiting the $HO_2\cdot$ oxidative pathway. Additionally, a further formation of sulfate in lifting air mass moderately enhanced ALWC aloft, leading to a low IEPOX reactive uptake on the particle surface, which would also limit the daytime $BSOA_I$ formation aloft. Unfortunately, our limited data does not allow us to peer into the fundamental mechanistic and kinetic details of this process. All these findings highlighted the complex and regional variability of the influences of NOx on BSOA formation. Over the past decade, atmospheric environment in China has undergone substantial changes due to unevenly implemented emission controls, resulting in much higher levels of $NO_2$ compared with $SO_2$ (Zheng et al., 2018). These alterations in pollutant emissions could substantially affect BSOA formation; Thus, there is an urgent need for long-

term characterization of BSOA to better assess its potential impacts on radiative

forcing, human health, and to fully understand the anthropogenic−biogenic

interactions.

**Data availability.** The data used in this study are freely available at

https://doi.org/10.5281/zenodo.15164940 (Wu, 2025). And Meteorological data and

hourly $PM_{2.5}$, $NO_2$, $O_3$ concentrations can be obtained from

https://doi.org/10.5281/zenodo.7413640 (Wu, 2022).

**Author contributions.** G.W. designed research and contributed analytic tools. C.W.,

C.C. and J.L. collected the samples. C.W. and Y.C. conducted the sample analysis.

C.W. and G.W. performed the data interpretation. C.W. wrote the paper. All authors

contributed to the paper with useful scientific discussions.

**Competing interests.** The authors declare no competing interest.

**Acknowledgements.** This work was financially supported by the National Natural

Science Foundation of China (grant no. 42477097, 42130704) and the National Key

Research and Development Program of China (grant no. 2023YFC3707401).

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

**Table 1** Summary of the average BSOA tracers at the two sampling sites during the
whole campaign.

| Compounds | Mountain foot | | | Mountainside | | |
|---|---|---|---|---|---|---|
| | Average | Daytime | nighttime | Average | Daytime | nighttime |
| **Isoprene derived SOA tracers** | | | | | | |
| 2-MGA[a] | 29±15 | 36±17 | 22±7 | 16±8 | 20±9 | 13±6 |
| **2-methyltetrols** | | | | | | |
| 2-Methylthreitol | 24±10 | 25±10 | 23±10 | 32±23 | 29±21 | 36±24 |
| 2-Methylerythritol | 50±2 | 52±23 | 48±21 | 67±46 | 57±42 | 76±49 |
| subtotal | 74±32 | 77±33 | 71±31 | 99±68 | 86±62 | 112±72 |
| **C$_5$-alkene triols** | | | | | | |
| cis-2-Me-1,3,4-THB[b] | 20±13 | 20±13 | 20±13 | 22±18 | 19±15 | 24±20 |
| 3-Me-2,3,4-THB[c] | 23±15 | 22±14 | 23±16 | 23±19 | 21±17 | 26±21 |
| trans-2-Me-1,3,4-THB[d] | 31±19 | 30±19 | 31±19 | 30±22 | 26±19 | 33±25 |
| subtotal | 74±46 | 73±45 | 74±48 | 75±59 | 66±51 | 83±66 |
| **3-MeTHF-3,4-diols** | | | | | | |
| trans-3-Me-THF-diol[e] | 3±1 | 3±1 | 3±1 | 3±2 | 3±2 | 3±2 |
| cis-3-Me-THF-diol[f] | 4±2 | 4±2 | 4±2 | 4.7±3.7 | 4±3 | 5±4 |
| subtotal | 7±3 | 7±3 | 7±3 | 8±6 | 8±5 | 8±6 |
| Total | 182±81 | 191±83 | 173±77 | 197±126 | 178±114 | 216±135 |
| **α/β-pinene derived SOA tracers** | | | | | | |
| cis-pinonic acid | 9±4 | 9±4 | 9±4 | 4±2 | 4±2 | 4±3 |
| pinic acid | 5±2 | 6±2 | 5±2 | 3±2 | 4±2 | 3±2 |
| MBTCA[g] | 26±14 | 30±14 | 23±14 | 19±13 | 21±14 | 18±12 |
| 3-HGA[h] | 12±6 | 14±6 | 11±6 | 6±4 | 7±5 | 4±3 |
| Total | 52±23 | 59±22 | 46±22 | 32±20 | 36±21 | 28±18 |
| **β-caryophyllene derived SOA tracer** | | | | | | |
| β-Caryophyllinic acid | 35±19 | 35±17 | 34±20 | 56±40 | 70±46 | 42±28 |

[a]2-MGA: 2-methylglyceric acid;
[b]cis-2-Me-1,3,4-THB: cis-2-Methyl-1,3,4-trihydroxy-1-butene;
[c]3-Me-2,3,4-THB: 3-Methyl-2,3,4-trihydroxy-1-butene;
[d]trans-2-Me-1,3,4-THB: trans-2-Methyl-1,3,4-trihydroxy-1-butene;
[e]trans-3-Me-TH-diol: trans-3-Methyltetrahydrofuran-3,4-diol;
[f]cis-3-Me-TH-diol: cis-3-Methyltetrahydrofuran-3,4-diol;
[g]MBTCA: 3-methyl-1,2,3-butanetricarboxylic acid;
[h]3-HGA: 3-Hydorxyglutaric acid

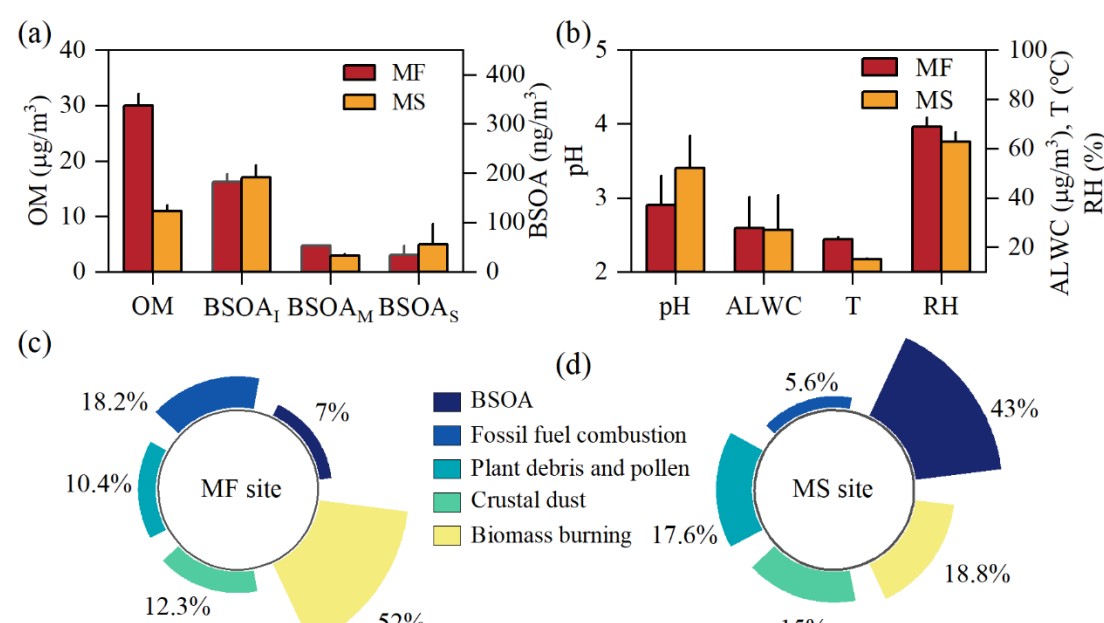

**Figure 1** Comparisons upon chemical composition **(a)**, meteorological conditions **(b)**, and sources for OM **(c and d)** among two sampling sites. (OM concentration is converted by the OM/OC ratio measured in our previous (Wu et al., 2024); The standard deviations of all species except BSOAs in Figures (a) and (b) were reduced by a factor of five)

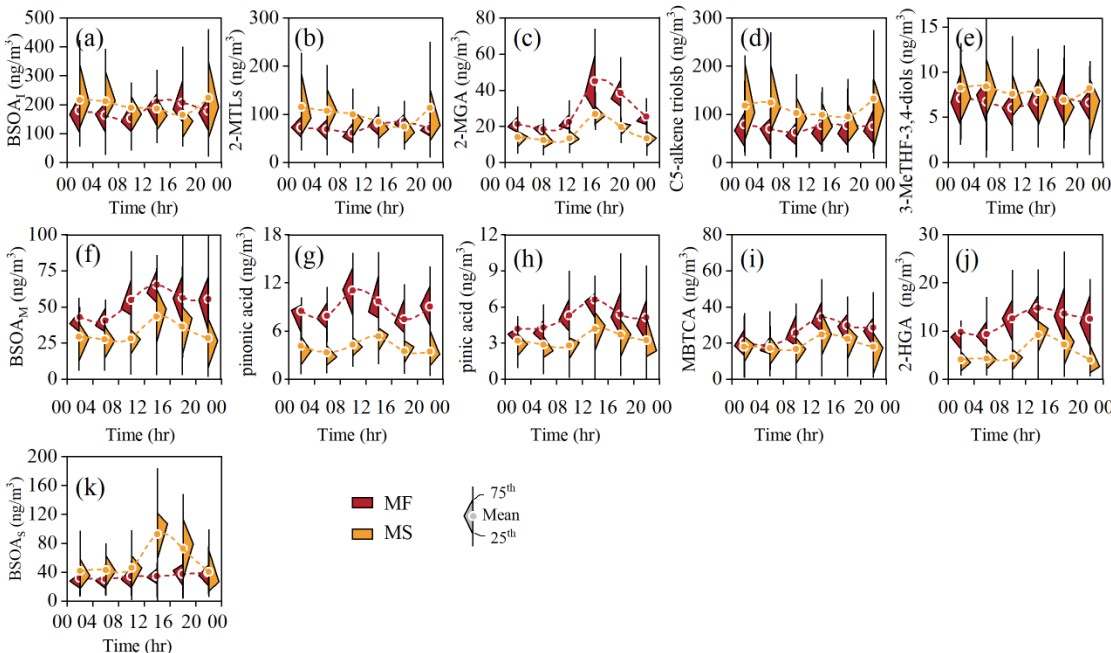

**Figure 2** Diurnal variations in BOSA tracers among both sites. **(a-e)** isoprene-derived SOA tracer; **(f-j)** monoterpenes-derived SOA tracer; **(k)** β-caryophyllene -derived SOA tracer.

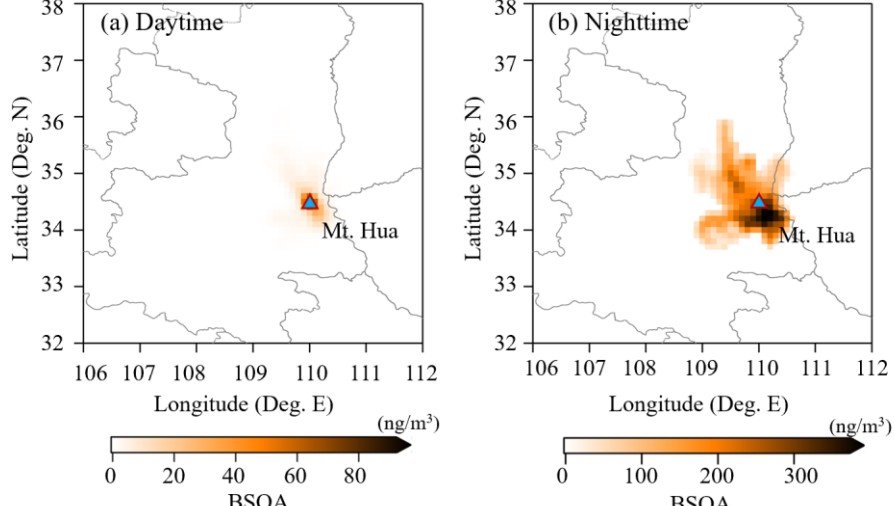

**Figure 3** A concentration-weighted trajectory (CWT) analysis for $BSOA_I$ at MS site.

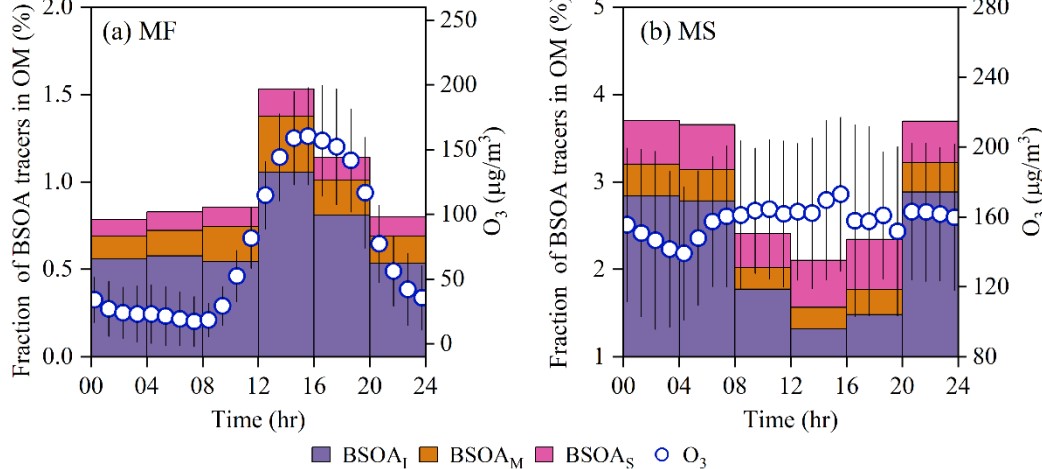

**Figure 4** Diurnal cycles of mass fraction of BSOA tracers in OM and $O_3$ at both sampling sites.

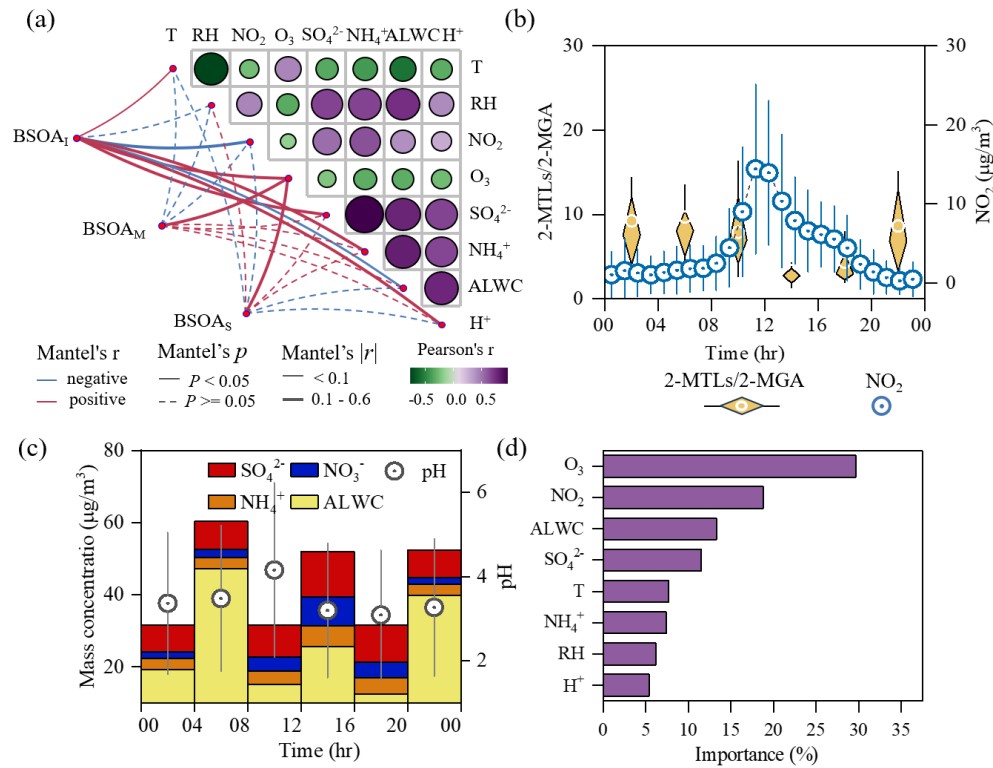


**Figure 5** Formation of daytime isoprene-derived SOA in upper troposphere. **(a)**
Mantel test between BSOA tracers and potential influencing factors at MS site; **(b**
**and c)** Diurnal variations 2-MTLs/2-MGA ratio, pH, and the concentration of $NO_2$,
SNA, ALWC at MS site; **(d)** Importance assessment for the key factors affecting the
daytime isoprene-derived SOA at the MS site.



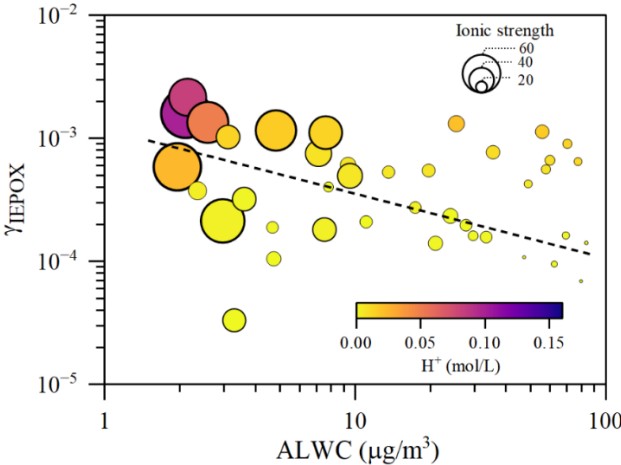


**Figure 6** Reactive uptake ($\gamma_{IEPOX}$) as a function of ALWC during the daytime at MS
site.