# Peer review of "Organic Aerosol Formation in the Atmosphere of Mt. Hua,"

_EGUsphere, 2025_

## Author Comment (AC2)

Dear ACP editor:

After reading the comments from you and the reviewers, we have carefully revised our manuscript. Our responses to the comments are itemized below.

Anything for our paper, please feel free to contact Prof. Gehui Wang via ghwang@geo.ecnu.edu.cn.

All the best

Can Wu
On behalf of Prof. Gehui Wang
Jun 23, 2025

Reviewer(s)' Comments to Author:

**Reviewer 1**

General comments: This manuscript discusses the influence of anthropogenic emissions on biogenic secondary organic aerosol formation in the upper boundary layer. The dataset includes samples collected at the mountain foot and mountain side of Mt. Hua China. They find that NOx impacts isoprene-derived BSOA formation at the mountain side and observe an inverse diurnal trend for isoprene-derived BSOA at the mountain foot and mountain side. Overall, the dataset is comprehensive and provides a valuable addition to the scientific community. I believe this manuscript is publishable after major revisions are conducted, as described below.

**Reply**: We thank the reviewer's valuable comments. We have thoroughly revised the manuscript accordingly and believe these revisions significantly strengthen our work. All modifications in the revised version are highlighted in yellow. Please see details below.

**Specific comments**:

(1) I would recommend that the authors review the more recent literature on isoprene-derived SOA formation. Specifically the articles Riva et al. 2019 (DOI: 10.1021/acs.est.9b01019) Lei et al. 2019 (https://doi.org/10.1021/acs.est.2c01579), Zhang et al. 2018 (DOI: 10.1021/acs.estlett.8b00044), and Cooke et al. (https://doi.org/10.1021/acs.est.3c10851). These experimental and modeling studies demonstrate that iSOA formation is more complex than what is described in the present article. For example, in lines 83 – 84 the authors that that sulfate would increase SOA

formation. However, the previous articles demonstrate that this is not always true as core/shell morphology can form and limit further reactive uptake reactions and highly acidic sulfuric acid particles can limit the formation of organosulfates due to the poor nucleophilicity of bisulfate. I recommend the authors consider revising to include these details.

**Reply**: Thanks for your advice. These references have provided valuable insights, enabling us to develop a more comprehensive understanding of BSOA formation mechanisms. We have incorporated these works into the introduction, and provided additional discussions regarding the influence of sulfate on BSOA formation as the follows. Please see page 3, line 81~91; and page 21, line 468~480.

*"For example, sulfate, acting as an effective nucleophile, greatly promotes ring-opening reactions of isoprene epoxydiols (IEPOX) and subsequent SOA formation, particularly organosulfates and corresponding oligomers (Cooke et al., 2024a; Xu et al., 2015); These products contribute significantly to IEPOX-driven SO, and can alter aerosol physicochemical properties (e.g., phase state, viscosity and morphology) (Lei et al., 2022; Riva et al., 2019), thereby kinetically mediating the reactive uptake of IEPOX, as weel as the following SOA yield and evolution (Drozd et al., 2013; Zhang et al., 2018; Armstrong et al., 2022). Furthermore, a recent laboratory study demonstrates that sulfate/bisulfate equilibrium also plays a critical role in BSOA formation, especially under highly acidic conditions (Cooke et al., 2024b)."*

*"The $SO_4^{2-}/HSO_4^-$ equilibrium is crucial for $BSOA_1$ yield under a highly acidic condition (Cooke et al., 2024b); While, the chemical form of inorganic S(VI) also shifts from $HSO_4^-$ at MF site to $SO_4^{2-}$ at MS site during vertical transport, which would facilitate $BSOA_1$ formation, as the nucleophilicity of $SO_4^{2-}$ is two orders of magnitude higher than that of $HSO_4^-$ (Aoki et al., 2020). As is well-known, sulfate can modulate the $H^+$ level, acting as a more efficient proton donor to catalyze IEPOX ring opening and isoprene ozonolysis; Thus, $H^+$ concentration predicted by the thermodynamic model ISORROPIA II positively correlates with $BSOA_1$ (r=0.37, P<0.05, Figure 5(a)). But a nonlinear relationship between the yields of methyltetrol sulfates/ 2-MTLs and under low pH (<3) condition is revealed in the chamber study (Cooke et al., 2024b); Based on our previous study, the average daytime pH was 3.4±2.2 at MS site, indicating that there are other factors perturbing BSOA formation at Mt. Hua, warranting further investigation."*

**Comment**

(2) Please include the pore size and manufacturer for the quartz filters used in collection.

It would be beneficial for the reader if you could describe how you determined the mass concentration for the organic matter as well as for the individual species in the methods section. Did you have calibration standards for your species of interest?

**Reply:** Following reviewer's advice, we have provided more details about the quartz filters, including the manufacturer and retention efficiency. The quantitative method for individual BSOA tracers and the recoveries of the standard solutions solutions were also presented in section 2.2. Due to the commercial unavailability of a subset of authentic standards, the quantification of target compounds relied on the surrogate standards, expect for cis-pinonic; This approach is also applied in other similar studies (Zhu et al., 2016; Kang et al., 2018; Wang et al., 2022). See page 7, line 149~152; and page 8, line 175~187.

*"All the aerosol samples were collected on pre-combusted (450 ℃ for 6 hr) quartz filters (Whatman 1851-865), which has a retention efficiency of >99.995% for DOP particles at 0.3 μm; After collection, the filter samples would be stored in a freezer (< −18 ℃) prior to chemical analysis."*

*"All BSOA tracers were individually identified by comparing mass spectra against authentic standards and literature data. Whereas, due to the commercial unavailability of a subset of authentic standards, the quantification of target compounds relied on the surrogate standards, expect for cis-pinonic. Specifically, the erythritol was applied to determine the 2-methyltetrols, $C_5$-alkene triols and 3-MeTHF-3,4-diols; The quantification of 2-methylglyceric acid, 3-hydroxyglutaric acid, 3-methyl-1,2,3-butanetricarboxylic acid and β-caryophyllinic acid was performed using glyceric acid, tartaric acid, suberic acid and cis-pinic acid, respectively. This approach was also applied in other similar studies (Li et al., 2013; Zhu et al., 2016). The standards solutions were spiked onto the blank filters to assess to the recoveries, which ranged from 82% to 105%. Field blank filters were also analyzed following the same analytical procedure as the ambient samples, with no target compounds detected in these blanks."*

**Comment**

(3) It would be helpful for the reader to include the Q values (Q $_{true}$, Q $_{robust}$, and their ratio) in section 2.4, particularly for others in the field who would like to conduct a similar PMF analysis of their data set.

**Reply:** As suggested, we have provided the Q values of interest to reviewers and readers, i.e, $Q_{True}$, $Q_{Robust}$ and $Q/Q_{expected}$ are 2362.8, 2365.6 and 0.86, respectively. These values are detailed in page 9, line 209~211; and page 10, line 216~220.

**Comment**

(4) Throughout the manuscript, there is consistent error in presenting the uncertainties. For example in lines 199 : "30.0±10.4". Uncertainties should only be provided to the last significant digit. Therefore, if your range is ±10, the technically correct way to present this would be "30±10".

**Reply:** Following the reviewer's suggestion, we have meticulously reviewed and modified the data point by point throughout the manuscript.

**Comment**

(5) A citation is needed substantiating the claim that hopanes are a known tracer for combustion sources in lines 202-203

**Reply:** Suggestion taken. The appropriate reference has been cited here. See page 11, line 247.

**Comment**

(6) In Figure 1, it is interesting to me that the mass concentrations of BSOA for I, M, and S, seem to be overall similar between MF and MS; yet, the percentages of sources are so different. Could the authors comment on this? I think a reader could have a similar curiosity.

**Reply:** Thanks for reviewer's advice. The concentration of $BSOA_M$ was MF: 52±23 ng/m$^3$ at MF site and 32±20 ng/m$^3$ at MS site, respectively; Thus, the data presented in the original Figure 1 was incorrect due to our oversight, and we have modified it in the manuscript. We think that a larger axis range in the figure might have made the differences in BSOA concentrations appear visually insignificant; While the spatial differences for BSOA tracers are indeed highly significant, particularly for $BSOA_M$ and $BSOA_S$ (Figure 1). Additionally, the source apportionment results in this study, which focused on organic matter (OM) rather than BSOA, are reasonable and consistent with our previous findings (Wu et al., 2022; Wu et al., 2024). Specifically, the MS site differs from the MF site because its steep terrain limits anthropogenic influence. Consequently, biogenic primary and secondary sources dominate the OM at the MS site, contrasting with the surface dominance of anthropogenic sources typically observed at the MF site.

[Figure]

**Figure 1** Spatial difference for BSOA tracers.

**Comment**

(7) In lines 230 – 233 and 264-266, the authors may be interested to know that their MF mass concentration of iSOA products, and specifically 2-MTLs, is also similar to that recently measured in two regions of Mexico City (Cooke et al. 2024 https://doi.org/10.1021/acsestair.4c00048)

**Reply:** Thank you for the valuable work that supported our observations; we have included the citation in the revised version. Please see page 14, line 310.

**Comment**

(8) A citation is needed in line 255 substantiating the claim that isoprene emissions are temperature-driven

**Reply:** The citations have been added following your advice. Please see page 13, line 300.

**Comment**

(9) Lines 260 – 264, describe "CWT analysis." Please define the acronym CWT for the reader and additionally provide details on this analysis in your methods section. As it is currently written, it is not clear to the reader what this refers to or how the data in this figure was generated.

**Reply:** Suggestion taken. More details for the concentration-weighted trajectory (CWT) analysis were provided in the method section (i.e., 2.5 section) as follows. See page 10, line 221~235.

*"To identify the potential spatial sources of the high BSOA on Mt. Hua, a CWT analysis was conducted using the BSOA concentrations and air mass backward trajectories. The 12-hr backward trajectories at a height of 1100 m were calculated by Hybrid-Single Particle Lagrangian Integrated Trajectory (HYSPLIT) model on an hourly basis*

*throughout the campaign. Due to the relatively low temporal resolution of filter samples, the averaged BSOA concentration was assigned to match all the trajectories corresponding to each sample. And the CWT analysis herein was subsequently performed using the Igor-based tool using the Igor-based tool (i.e., ZeFir v3.70); More details for the protocol can can be found in Petit et al. (2017). Briefly, a grid with 0.25° × 0.25° cell size was created to cover all the trajectories; Each grid cell would be assigned a weighted concentration, which obtained from the averaged BSOA concentration that have associated trajectories crossing the grid cell. A high value in a gird cell indicates that the air parcels passing through it were associated with high concentration at the receptor site. "*

**Comment**

(10) A citation is needed in line 290-293 for the studies that have discussed the formation mechanism of 2-MTLs and $C_5$-alkene triols

**Reply:** Suggestion taken. The appropriate references are included here. Please see page 15, line 341 and 343.

**Comment**

(11) Have the authors found the presence of any BSOA(I) oxidation products? It would be interesting to see if there is a difference in these products at the MF vs. MS. See Armstrong et al. 2022 https://doi.org/10.1021/acs.est.2c03200 and more recently Yan et al. 2025 https://doi.org/10.1021/acs.jpca.4c08082

**Reply:** Only eight molecules of isoprene oxidation products (e.g., methyltetrols) were detected in this study. However, the dimers and oligomers of organosulfate formed during the further oxidation of epoxydiol-derived secondary organic aerosol were not available here. Therefore, a deeper discussion of the spatial differences in $BSOA_I$ oxidation products was not conducted here. Even so, these important findings from the referenced study have been incorporated into the revised manuscript. Please see page4, line 88; and page 14, line 434.

**Comment**

(12) Lines 367 – 368, it looks to me like only BSOA(I) correlates positively with ambient temperature and that BSOA(M) and BSOA(S) negatively correlate with temperature.

**Reply:** Sorry for our inaccuracy expression. We have undertaken a reanalysis and subsequently revised the relevant text to ensure more accuracy. Please see page 19, line 418~425.

**Comment**

(13) Please provide references in line 379 – 382 substantiating the claim that ALWC, aerosol acidity, and gas-particle partitioning would influence BSOA formation

**Reply:** Suggestion taken. The relevant references have been provided. Please see page 19, line 434.

**Comment**

(14) Please provide reference is lines 391 – 392 substantiating the claim about high NOx/low NOx SOA pathways

**Reply:** Suggestion taken. See page 20, line 448.

**Comment**

(15) Lines 402 – 404, unclear to me how this shows that sulfate has a statistically significant relationship with BSOAI. Incorrect figure referenced? Not sure what this is meant to refer to.

**Reply:** Sorry for our carelessness. The referenced figure here is Figure 5(a)

**Comment**

(16) In this and the following sections, please see the above comment about more recent literature on the role of sulfate, morphology, and acidity in iSOA formation. The literature cited here is not up to date, and thus the interpretation of the measurements is not complete.

**Reply:** Suggestion taken. Additional discussions about the influences of sulfate on $BSOA_I$ formation have been conducted both in the induction and this section according to the findings revealed by above valuable literatures. Please see page 3, line 81~91; and page 21, line 468~480.

**Comment**

(17) Lines 417 – 418, again it is unclear to me how Figure 5c supports the claim in the text. Please reference Figure 6 properly in the text in paragraph starting in lines 417

**Rely:** In our previous study, the mass fraction of sulfate at MS site moderately enhanced compared with MF site (Figure 2), suggesting a further formation of sulfate in a lifting air mass; Concurrently, chemical form of sulfate also shifted from $NH_4HSO_4$ at MF site to $(NH_4)_2SO_4$ at MS site. In contrast, nitrate exhibited a decreasing trend due to volatilization. Thus, we infer that the further sulfate formation would promote an increase in ALWC at MS site, as substantiated by a positive correlation between sulfate

and ALWC. These discussion and evidences were also provided in the revised manuscript, please see page 21, line 461~464; and page 22, line 481~485.

[Figure]

**Figure 2** The comparison of proportions in daytime SNA between the two Sampling Sites **(a)**. ALWC as a function of daytime sulfate at MS site **(b)**.

**Comment**

(18) Line 439, please explain to the reader what is meant by "RF analysis" and include details on this in either the methods section or supporting information Technical corrections:

**Reply:** Suggestion taken. More details for random forest analysis (RF) and Mantel's test have been provided in the supporting materials as follows.

*"Randow forest (RF) was initially introduced by Breiman (2001) to serve as a powerful tool for regression and prediction with high accuracy; and it has been widely applied in environmental field (e.g., pollutants prediction), even the data have complex nonlinear relationships and interactions. In this study, a regression model based on RF algorithm was built to reveal the key factors influencing the BSOA formation at MS site. The potential factors, including $O_3$, $NO_2$, $SO_4^{2-}$, $NH_4^+$, ALWC, $H^+$, RH and T, were regarded as explanatory variables for the daytime isoprene-derived SOA prediction. For robust model development and validation, 70% of the original input data were randomly allocated to build the RF model (i.e., training dataset), and the remaining 30% served as the testing dataset for evaluating the model performance. The RF model is user-friendly with only two critical parameters constantly being optimized, namely the number of trees grown ($n_{tree}$) and the number of variables split at each node ($m_{try}$). By systematically comparing the results of different parameters settings, we identified that setting $n_{tree}$ to 100 and $m_{try}$ to 4 can provide the optimal prediction accuracy, which can*

*be established by the statistical metrics as shown in Table S1. From Table S1, we can note that a well performance of the model in explaining the importance of these factors to daytime BSOA$_I$, with a robust correlation between predicted and observed BSOA ($R^2$=0.8). Additionally, the mantel test, as a statistical tool initiated by Mantel (1967), is commonly used for investigating relationships among two matrices. To elucidate the relationships between three types of BSOA tracers and potential influencing factors, a Mantel test was conducted using a network platform (https://www.omicstudio.cn/tool, last accessed: Jan. 10$^{th}$, 2025)."*

**Comment**

(19) Line 108 "observations studies" to "observational studies"

**Reply:** Suggestion taken. See page 5, line 120.

**Comment**

(20) Line 115 "dramatical" to "dramatic"

**Reply:** Suggestion taken. See page 6, line 127.

**Comment**

(21) Lines 213-214 "believed to" to "believed to be"

**Reply:** Sorry for our oversight. The issue has been corrected. See page 11, line 258.

**Comment**

(22) Lines 241 -242, should this say "more densely distributed"?

**Reply:** Actually, the coniferous plants are more abundant at the MS site than that at the MF site. Thus, the high BSOA$_M$ levels observed at MF are likely due to intensive biomass burning emissions.

**Comment**

(23) In Figure 5, "pearson's" has a red line underneath

**Reply:** The underline has been removed in the revised Figure 5.

**Comment**

(24) Lines 367, "Figure 6(a) to Figure 5a?

**Reply:** Suggestion taken. See page 19, line 418.

**Comment**

(25) Line 367, BOSA to BSOA

**Reply:** Suggestion taken. See page 19, line 418.

**Comment**

(26) Line 387 – Figure 6(a) to Figure 5a?

**Reply:** Suggestion taken. See page 20, line 442.

**Reviewer 2**

**General comments**

This manuscript reports the field measurements of biogenic SOA tracers and anthropogenic emissions at two sampling sites in the Mount Hua region at different altitudes. Overall, while this study presents some interesting results, my primary concern is the lack of proper references in the discussion, which could impact the data interpretation. There are many typos and grammatical errors throughout the manuscript. I suggest the authors carefully correct these errors to improve the clarity of presented results. Below I have a few specific questions for the authors to consider.

**Reply**: We sincerely appreciate the reviewer's valuable comments. We have carefully addressed all the concerns and thoroughly revised the manuscript accordingly. See details below.

**Specific comments**

(1) Lines 147-155: Please provide information regarding the extraction efficiencies for major BSOA tracers using the given analytical method.

**Reply:** Suggestion taken. The recovery and quantitative method for target compounds were additionally provided in 2.2 section. Please see page 8, line 175~187.

*"All BSOA tracers were individually identified by comparing mass spectra against authentic standards and literature data. Whereas, due to the commercial unavailability of a subset of authentic standards, the quantification of target compounds relied on the surrogate standards, expect for cis-pinonic. Specifically, the erythritol was applied to determine the 2-methyltetrols, $C_5$-alkene triols and 3-MeTHF-3,4-diols; The quantification of 2-methylglyceric acid, 3-hydroxyglutaric acid, 3-methyl-1,2,3-butanetricarboxylic acid and β-caryophyllinic acid was performed using glyceric acid, tartaric acid, suberic acid and cis-pinic acid, respectively. This approach was also applied in other similar studies (Li et al., 2013; Zhu et al., 2016). The standards solutions were spiked onto the blank filters to assess to the recoveries, which ranged*

*from 82% to 105%. Field blank filters were also analyzed following the same analytical*
*procedure as the ambient samples, with no target compounds detected in these blanks."*

**Comment**

(2) Lines 155-161: Could the authors please verify the unit of "Ml" mentioned here?
Do they mean milliliters (mL) or microliters (μL)?

**Reply:** Sorry for our carelessness. It should be microliters (μL) here, and we have
corrected it. See page 7, line 170~174.

**Comment**

(3) Lines 207-210: "Nonetheless, an inverse diel pattern was observed at high elevation
site with a daily OM peak at the moment (12:00~16:00) of strong photochemical
activity, suggesting that OM components aloft was probably driven by photochemistry."
Does this statement refer to Figure S1(b)? Please specify.

**Reply:** This statement accurately reflects the content depicted in Figure S1(b), and we
have annotated it accordingly. See page 11, line 250~253.

**Comment**

(4) Line 219: I would suggest changing "septuple" to "7-fold" here and using similar
expressions throughout the manuscript if you intend to compare the fold changes.
Similarly, I would suggest using "surface OM" and "OM aloft" throughout the
manuscript if you intend to compare OM at different altitudes.

**Reply:** Suggestion taken. We have modified it throughout the manuscript.

**Comment**

(5) Line 260: Please define "CWT" for its first use in the main text.

**Reply:** Following your suggestion, we have defined the acronym CWT at its first
occurrence, and provided more details about this analysis in the method section as
follows. Please see page 10, line 221~235.

*"To identify the potential spatial sources of the high BSOA on Mt. Hua, a CWT analysis*
*was conducted using the BSOA concentrations and air mass backward trajectories. The*
*12-hr backward trajectories at a height of 1100 m were calculated by Hybrid-Single*
*Particle Lagrangian Integrated Trajectory (HYSPLIT) model on an hourly basis*
*throughout the campaign. Due to the relatively low temporal resolution of filter samples,*
*the averaged BSOA concentration was assigned to match all the trajectories*
*corresponding to each sample. And the CWT analysis herein was subsequently*

*performed using the Igor-based tool using the Igor-based tool (i.e., ZeFir v3.70); More details for the protocol can can be found in Petit et al. (2017). Briefly, a grid with 0.25° × 0.25° cell size was created to cover all the trajectories; Each grid cell would be assigned a weighted concentration, which obtained from the averaged BSOA concentration that have associated trajectories crossing the grid cell. A high value in a gird cell indicates that the air parcels passing through it were associated with high concentration at the receptor site."*

**Comment**

(6) Lines 278-280: "This finding was consistent with the laboratory measurements of 2-MGA being derived from oxidization of isoprene with high NOx load (Wang et al., 2008)." The "Wang et al., 2008" cited in the manuscript was not a laboratory study. Please provide correct references to support this statement.

**Reply:** Suggestion taken. The appropriate references were provided in the revision version. Please see page 15, line 329.

**Comment**

(7) Lines 289-293: "This was probably owing to that C5-alkene triols and 3-MeTHF-3,4-diols were mainly formed via acid-catalyzed intermolecular rearrangement reactions of epoxy-diols, whereas 2-MTLs are likely to the result from nucleophilic addition of water to the ring opening of epoxy-diols." These statements require appropriate references. Regarding 2-MTLs, their formation has been linked to acid-catalyzed reactive uptake of isoprene epoxydiols. Recent field studies also suggest that 2-methyltetrols could be produced through biological processes within the plants and released into the atmosphere as primary emissions. Therefore, 2-MTLs can potentially originate from multiple sources (both primary and secondary) and via different mechanisms, all of which should be considered.

Lin, et al. "Isoprene epoxydiols as precursors to secondary organic aerosol formation: acid-catalyzed reactive uptake studies with authentic compounds." Environmental Science & Technology 46.1 (2012): 250-258.

Ye, et al. "Near-canopy horizontal concentration heterogeneity of semivolatile oxygenated organic compounds and implications for 2-methyltetrols primary emissions." Environmental Science: Atmospheres 1.1 (2021): 8-20.

**Reply:** Following the suggestion, the appropriate references have been incorporated into the revised manuscript. As shown in Figure 3, 2-MTLs are not correlated with $C_{29}$-alkane, a compound primarily released by vegetation (Wang and Kawamura, 2005);

This finding indicates that 2-MTLs at both sampling sites mainly originated from secondary formation, rather than being directly emitted by vegetation. These discussions have been added in the manuscript, please see page 14. line 312~317.

[Figure]

**Figure 3** Linear correlations of 2-MTLs with $C_{29}$ at both sampling sites.

**Comment**

(8) Lines 330-332: "As a typical and abundant sesquiterpene, β-caryophyllene has been widely studied due to its high reactivity and significant aerosol formation potential; and it can be oxidized into β-caryophyllinic acid for via ozonolysis/photooxidation." Please provide appropriate references to support this statement.

**Reply:** As requested, the supporting references have been included in the revised manuscript. Please see page 17, line 383.

**Comment**

(9) Line 347: typo on "Ox load"

**Reply:** Suggestion taken. See page 18, line 397.

**Comment**

(10) Lines 364-365: Details for the mantel test and random forest analysis mentioned here should be included in the method section or the supplementary materials.

**Reply:** Suggestion taken. More descriptions about the mantel test and random forest analysis have added in the supplementary materials (Text S1) as follows.

*"Randow forest (RF) was initially introduced by Breiman (2001) to serve as a powerful tool for regression and prediction with high accuracy; and it has been widely applied in environmental field (e.g., pollutants prediction), even the data have complex nonlinear relationships and interactions. In this study, a regression model based on RF*

*algorithm was built to reveal the key factors influencing the BSOA formation at MS site. The potential factors, including $O_3$, $NO_2$, $SO_4^{2-}$, $NH_4^+$, ALWC, $H^+$, RH and T, were regarded as explanatory variables for the daytime isoprene-derived SOA prediction. For robust model development and validation, 70% of the original input data were randomly allocated to build the RF model (i.e., training dataset), and the remaining 30% served as the testing dataset for evaluating the model performance. The RF model is user-friendly with only two critical parameters constantly being optimized, namely the number of trees grown ($n_{tree}$) and the number of variables split at each node ($m_{try}$). By systematically comparing the results of different parameters settings, we identified that setting $n_{tree}$ to 100 and $m_{try}$ to 4 can provide the optimal prediction accuracy, which can be established by the statistical metrics as shown in Table S1. From Table S1, we can note that a well performance of the model in explaining the importance of these factors to daytime $BSOA_I$, with a robust correlation between predicted and observed BSOA ($R^2=0.8$). Additionally, the mantel test, as a statistical tool initiated by Mantel (1967), is commonly used for investigating relationships among two matrices. To elucidate the relationships between three types of BSOA tracers and potential influencing factors, a Mantel test was conducted using a network platform ([https://www.omicstudio.cn/tool](https://www.omicstudio.cn/tool), last accessed: Jan. $10^{th}$, 2025)."*

**Comment**

(11) Line 370: typo on "rom"

**Reply:** Sorry for our carelessness. We have corrected it.

**Comment**

(12) Lines 376-377: Please specify the significance level.

**Reply:** Suggestion taken. See page 19, line 429.

**Comment**

(13) Lines 384-385: "Evidences from field and modeling studies indicates that biogenic SOA yield is enhanced in presence of elevated NOx level (Xu et al., 2015; Shrivastava et al., 2017)." This statement is overly simplified and lacks sufficient detail regarding the role of NOx in biogenic SOA formation. The authors should more carefully discuss how NOx affect the fate of $RO_2$ radicals in the presence and absence of NOx, and how the subsequent reaction products contribute to SOA formation and yields.

**Reply:** Following your advice, additional discussions on how NOx affects the fate of $RO_2$ radicals, chemical composition of products, and BSOA yield have been added to the introduction section as follows. We also revised the inappropriate statement at the

beginning of Section 3.3.2. See page 4, line 91~99, and page 20, line 437~440.

*"NOx as an important driver for BSOA formation could alter the fate of organo-peroxy radicals (RO$_2$·), and subsequently affect the yield and chemical composition of BVOC-oxidized products by changing oxidation pathways (Lin et al., 2013; Pye et al., 2010). Specifically, RO$_2$·+HO$_2$· reactions under low NOx conditions would predominantly yield low-volatility hydroperoxide species; Conversely, the reaction of NO with RO$_2$·in the high NOx regime will produce organonitrates and alkoxy radicals (RO·) that can fragment into more volatile products. Whereas, the impact of NOx on BSOA yield is nonlinear as explored by laboratory studies (Xu et al., 2014; Kroll et al., 2006)."*

*"Evidence from laboratory studies referred to a nonlinear dependence of BSOA yield on NOx load, wherein the yield increases with rising NOx levels under low-NOx conditions, but exhibits a decreasing trend as NOx levels rise under high-NOx conditions (Xu et al., 2014; Xu et al., 2025)."*

**Comment**

(14) Lines 387 and 389: Figure 6(a) and Figure 6(b) here should be corrected with Figure 5(a) and Figure 5(b).

**Reply:** Suggestion taken. We have corrected it in the revised manuscript. See page 20, line 442~445.

**Comment**

(15) Lines 391-393: "As is well known, isoprene oxidization primarily follows two pathways, i.e., HO$_2$ pathway forming 2-MTLs in NOx-limited conditions and NO/NO2 channel yielding 2-MGA in high-NOx scenarios." Please provide appropriate references to support this statement.

**Reply:** Suggestion taken. The references have been included in the revised version. See page 20, line 448.

**Comment**

(16) Line 403: typo on "filed"

**Reply:** Suggestion taken. We have modified it, see page 20, line 456.

**Reference**

[revised manuscript text omitted]